# Intent-DQD: An Intent-based and Annotation-free Method for Duplicate Question Detection in CQA Forums

**Yubo Shu[1], Hansu Gu[2], Peng Zhang[1]**[*]**, Tun Lu[1]**[*]**, Ning Gu[1]**

[1]Fudan University, Shanghai, China
[2]Seattle, United States
ybshu20@fudan.edu.cn, hansug@acm.org
{zhangpeng_, lutun, ninggu}@fudan.edu.cn

## Abstract

With the advent of large language models (LLMs), Community Question Answering (CQA) forums offer well-curated questions and answers that can be utilized for instruction-tuning, effectively training LLMs to be aligned with human intents. However, the issue of duplicate questions arises as the volume of content within CQA continues to grow, posing a threat to content quality. Recent research highlights the benefits of detecting and eliminating duplicate content. It not only enhances the LLMs' ability to generalize across diverse intents but also improves the efficiency of training data utilization while addressing concerns related to information leakage. However, existing methods for detecting duplicate questions in CQA typically rely on generic text-pair matching models, overlooking the intent behind the questions. In this paper, we propose a novel intent-based duplication detector named Intent-DQD that comprehensively leverages intent information to address the problem of duplicate question detection in CQA. Intent-DQD first leverages the characteristics in CQA forums and extracts training labels to recognize and match intents without human annotation. Intent-DQD then effectively aggregates intent-level relations and establishes question-level relations to enable intent-aware duplication detection. Experimental results on fifteen distinct domains from both CQADupStack and Stack Overflow datasets demonstrate the effectiveness of Intent-DQD. Reproducible codes and datasets will be released upon publication of the paper.

## 1 Introduction

With the rapid development of web technology, Community Question Answering (CQA) forums have emerged as crucial platforms for individuals to seek solutions to their problems. These forums serve as high-quality knowledge bases, offering well-curated questions and answers that can be effectively utilized for instruction-tuning for large language models (LLMs). Instruction tuning plays a vital role in the Reinforcement Learning from Human Feedback (RLHF) process employed in building systems like ChatGPT and Claude (Ouyang et al., 2022; Bai et al., 2022). The purpose of instruction tuning is to train LLMs to better follow human intents. For instance, when a human posts a question to an LLM, instruction tuning helps the LLM learn to provide an answer instead of merely imitating the act of asking a question. As the volume of content in CQA forums continues to increase, the issue of duplicate questions becomes more prevalent, thereby posing a significant challenge to content quality.

Recent research emphasizes the significance of detecting and removing duplicate content in the instruction-tuning of LLMs. It promotes generalization across diverse intents (Hernandez et al., 2022), reduces overlap between training and test sets (Lee et al., 2022), mitigates information leakage risks (Kandpal et al., 2022), enhances learning stability, and improves training data efficiency. Inspiringly, Zhou et al. (2023) discovers that even a carefully selected set of 1,000 unique questions can effectively train an LLM to follow intents and achieve competitive performance when compared to OpenAI's GPT3-DaVinci003 model.

Despite the impressive results of previous studies in detecting duplication Liang et al. (2019); Xia et al. (2021); Peng et al. (2022), these methods do not consider intents in question, which may lead to sub-optimal results when deduplicating content. Figure 1 shows two question pairs that illustrate the importance of considering intents in duplication detection for instruction-tuning. From a generic text-pair matching perspective, both the exemplar duplicate and neutral pairs are partially semantically matched. Despite being mismatched in the non-intent part, the first pair is labeled as duplica-

---
[*]Corresponding author

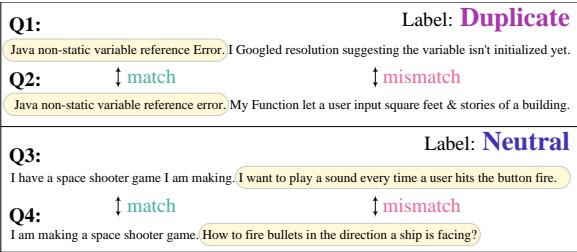

Figure 1: Example question pairs sampled from CQA forums. The highlights are the intents in questions. In the first pair, Q1 and Q2 are annotated as "Duplicate" by forum users. Q1 and Q2 have **matched intents** to solve a particular java error while having **mismatched backgrounds**. In the second pair, Q3 and Q4 are treated as neutral questions in the forum. Q3 and Q4 have **matched backgrounds** but **mismatched intents**.

tion due to the matched question intent. Meanwhile, the second pair only matches in non-intent content and thus should be treated as neutral.

Therefore, our goal is to highlight intents in duplicate question detection. However, this task remains challenging as follows: (1) Accurately recognizing intent from users' posts is difficult due to the high diversity of intent in different QA forums, especially without sufficient human annotation. (2) Determining whether two lexically different intents have the same semantics can be difficult because most annotations are at question-level instead of intent-level. (3) One question posted in online forums often carries more than one intent, and it is unclear how to handle multiple intents for duplicate question detection.

To address the above challenges, we first leverage the characteristics in CQA forums and extract abundant training labels to both recognize and match intents without human annotation. Based on matched intent pairs, we leverage supervised contrastive learning (Khosla et al., 2020) and properly learn intent representations in a low-dimensional space. Finally, we design a similarity matrix on the question pair level, based on inter and intra similarities between multiple intents for each question to detect duplicate questions.

The key contributions of this paper are summarized as follows:

- We leverage the features of intent in CQA forums and implement an effective intent recognizer without human annotation.

- We automatically match semantically identical intent pairs and separate non-identical ones through supervised contrastive learning.

- We propose a method to aggregate intent-level relations and obtain question-level relations to detect duplicate questions with multiple intents.

- The experimental results on two datasets covering fifteen domains demonstrate that our method can leverage question intent and achieve better performance over previous methods.

## 2 Annotation-free Label Collection

The primary challenge is the absence of labels for (1) intent recognition, which involves determining whether a sentence expresses an intent or not; (2) intent matching, which involves classifying whether the relationship between intents is matched or mismatched. In order to overcome this challenge, we introduce strategies for mining labels without human annotation by leveraging the characteristics of CQA forums.

### 2.1 Labels for Intent Recognition

The intent of a question refers to the user's goal or purpose in asking it. We have found that in most cases, the question title serves as a concise representation of what users are asking, which can be considered as intent. However, only using titles may lead to noise when titles are too short or do not provide enough information, such as "Order list" and "What should I do?". Inspired by previous research (Beyer et al., 2020), we additionally use regular expression patterns to match more accurate intents and eliminate overly generic intents only with pronouns, such as "how to do it/this/that". More information about used regex patterns is in Appendix A.1.

### 2.2 Labels for Intent Matching

After recognizing intents, there exist three types of links which can be used to derive valid labels for intent matching. Consider a common situation where a pair of questions is labeled as duplication in CQA forums, as shown in Figure 2, we can extract semantically matched intent pairs from inter, intra and mixed-links:

- **Intra-link** Inside one question, the intent in the title and the intent in the body can be linked. As the title and the body are expected to be consistent, their intents are likely semantically the same.

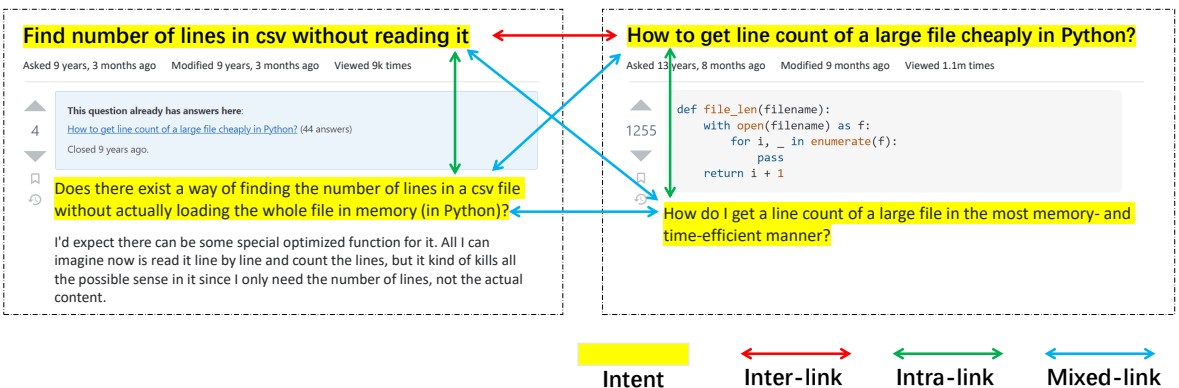

Figure 2: An example of mining matched intents through inter, intra, mixed-links in a pair of duplicate questions.

- **Inter-link** Between a pair of duplicate questions, intents in their titles can be duplicate as well.

- **Mixed-link** We can combine intra-links and inter-links to create mixed-links and further generate more training samples for intent matching.

Besides question pairs, it is worth noting that the intra-link can also work even within a single question. Similarly, we extract negative intent pairs from questions labeled as "related but not duplicate" in CQA forums by utilizing inter and mixed links. These negative intent pairs often exhibit lexical similarity but possess distinct semantic meanings. Consequently, pairs formed by these intents can serve as hard negative labels for training.

## 3   Method

After label collection, we proceed to present our method for detecting duplicate questions. Intent-DQD consists of three key components: intent recognition module, intent matching module, and duplicate question detection module. Figure 3 shows the overall structure of our method. In the intent recognition module, we exploit the inner features of intent sentences to augment recognition accuracy. In the intent matching module, we leverage supervised contrastive learning (Khosla et al., 2020) and properly learn intent representations in a low-dimensional space. In the duplicate question detection module, we design a similarity matrix on the question pair level to integrate inter and intra-similarities between multiple intents.

### 3.1   Intent Recognition

To ensure the accuracy of intent recognition, we incorporate three types of features in addition to transformer-based language models. We first focus on keywords that match tags in the QA forum. Matches such as "Tensorflow" or "PyTorch" indicate the mention of important concepts that are recognized by the QA forum, and are thus helpful for the model to learn if they are related to intent detection. We build a keyword vocabulary from all tags in the forum and use a binary flag to indicate whether the current word matches the tag. Additionally, considering the sentence "What is the difference between Concept A and Concept B?", we can classify the sentence as intent without knowing the specific concepts. So, we add part-of-speech (POS) information to help the model utilize the linguistic features of a sentence. Lastly, we use sentence position in the question as a feature because some users may prefer to put intents at the start or the end of a post. We implement all three features as extra embeddings added on top of existing text token embedding as illustrated in Figure 3(b).

### 3.2   Intent Matching

To distinguish matched or mismatched intents, we develop a dedicated intent matching layer based on contrastive learning. After the label collection process, we have collected both matched intents pairs (positive samples) and mismatched intent pairs (negative samples). We then feed these intent pairs to calculate the loss function of supervised contrastive learning. More specifically, $IntSet$ represents a dataset of intents, and $inent_k$ is the k-th intent in the dataset. Inside a training batch, we define: $P(k)$ contains positive intents

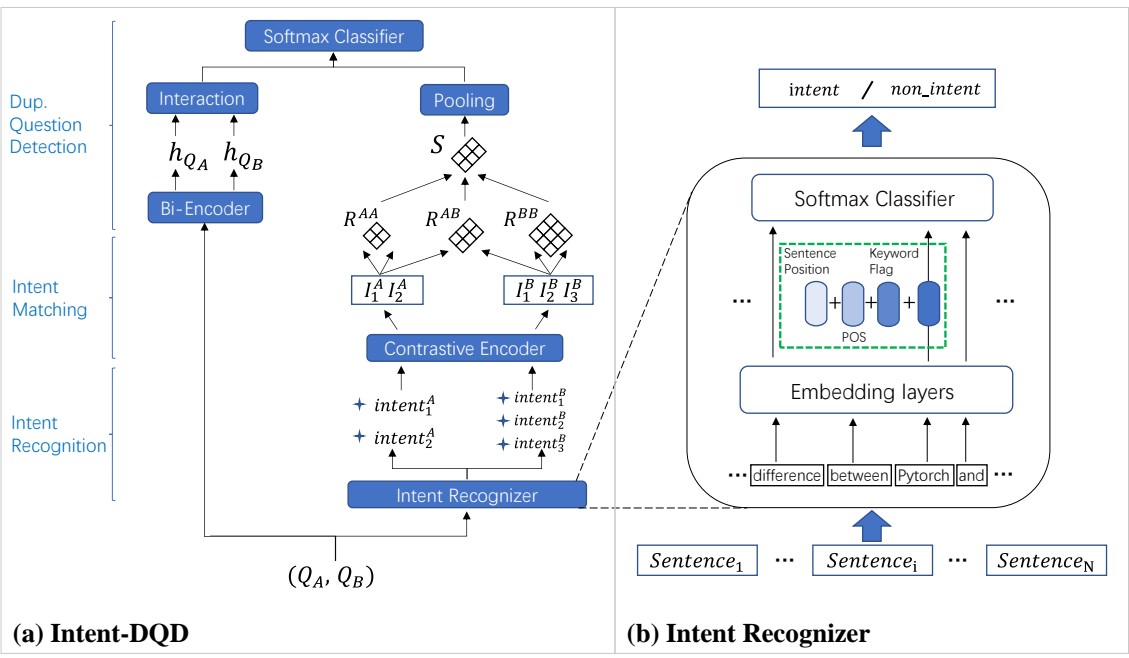

(a) Intent-DQD

(b) Intent Recognizer

Figure 3: The overall structure of our method. Part (a) on the left shows proposed intent-DQD processing question pairs through intent recognition, intent matching and duplicate question detection module. Part (b) on the right shows the details of the intent recognizer. Filtering by Intent prediction. After intent recognition, our method will select sentences predicted as the Intent label, passing them to the intent matching module.

of $inent_k$; and $N(k)$ contains negative intents of $inent_k$; $A(k) \equiv N(k) \cup P(k) \setminus inent_k$ is a union set excluding $inent_k$. For $inent_k$, the supervised contrastive loss $l$ is formulated as:

$$l = \frac{-1}{|P(k)|} \sum_{p \in P(k)} \log \frac{\exp(sim(I_k, I_p)/\tau)}{\sum_{a \in A(k)} \exp(sim(I_k, I_a)/\tau)}$$

where $I$ denotes the encoded vector of a intent, and $sim$ donates cosine similarity function and $\tau \in \mathbb{R}^+$ is a scalar temperature parameter. During the training process, optimizing the loss function for each intent in $IntSet$ will decrease the distance between positive intents in the embedding space and increase the distance between negative intents.

We use the intent embeddings from contrastive learning to match intents within a pair of questions. As shown in figure 3, given two questions $Q_A$ and $Q_B$, we encode the intents of $Q_A$ as a list of intent embeddings $[I_1^A, I_2^A, ..., I_M^A]$ and intents of $Q_B$ as $[I_1^B, I_2^B, ..., I_N^B]$. The $M$ and $N$ represent the number of intents in $Q_A$ and $Q_B$, respectively (e.g., $M = 2$ and $N = 3$ for the case in the figure). We match intents within each question and across questions, generating the following intent similarity matrices:

$$R^{AA} = [I_1^A, I_2^A, ..., I_M^A] \cdot [I_1^A, I_2^A, ..., I_M^A]^\mathsf{T}$$

$$R^{BB} = [I_1^B, I_2^B, ..., I_N^B] \cdot [I_1^B, I_2^B, ..., I_N^B]^\mathsf{T}$$

$$R^{AB} = [I_1^A, I_2^A, ..., I_M^A] \cdot [I_1^B, I_2^B, ..., I_N^B]^\mathsf{T}$$

The first matrix $R^{AA} \in \mathbb{R}^{M \times M}$ is used to match intents within $Q_A$. Similarly, the second $R^{BB} \in \mathbb{R}^{N \times N}$ is used to match intents within $Q_B$. The third $R^{AB} \in \mathbb{R}^{M \times N}$ is used to match intents between $Q_A$ and $Q_B$. The idea behind matching intent in this way is to provide a comprehensive view of question-level relation identification. In the next module, we will introduce how to aggregate these intent-level relations into question-level relations for duplication detection.

### 3.3 Duplicate Question Detection

It is worth noting that a question can have one or multiple intents. Therefore, in the duplicate question detection module, we combine the above matrices and calculate a holistic similarity matrix $S$ between $Q_A$ and $Q_B$. The shape of $S$ is $M * N$. The similarity score at the i-th row and j-th column, $S_{ij}$, represents not only an intent-level relation between $intent_i^A$ and $intent_j^B$, but also the degree to which this intent-level relation contributes to the question-level relation. For example, if $intent_i^A$ and $intent_j^B$ are semantically matched, its contribution degree to $S_{ij}$ depends on two situations: (1)

$intent_i^A$ is also matched with other intents in $Q_A$, indicating that $Q_A$ is a focused question, which makes the match between $intent_i^A$ and $inent_j^B$ more significant for duplication. (2) $intent_i^A$ is mismatched with other intents in $Q_A$, meaning that the match between $intent_i^A$ and $intent_j^B$ has a minor contribution for duplication. For this purpose, when calculating $S_{ij}$, we take into account the inner intent similarity matrices $R^{AA}$ and $R^{BB}$ with $R^{AB}$. The explicit formula of $S_{ij}$ is below:

$$S_{ij} = \frac{\sum\limits_{m=1}^{M} \sum\limits_{n=1}^{N} R_{im}^{AA} \cdot R_{mn}^{AB} \cdot R_{nj}^{BB}}{(\sum\limits_{m=1}^{M} R_{im}^{AA}) \cdot (\sum\limits_{n=1}^{N} R_{nj}^{BB})} \quad (1)$$

After calculating $S \in \mathbb{R}^{M \times N}$, as shown in Figure 3, we obtain the final feature $f$ by concatenating the max pooling of $S$ with the interaction of question embedding $h_A$ and $h_B$ from the bi-encoder:

$$f = [h_A : h_B : |h_A - h_B| : MaxPooling(S)]$$

We use a linear layer followed by a Softmax to make the final prediction:

$$\hat{y} = softmax(ReLU(Wf + b)),$$

where $\hat{y}$ is the prediction of probability in $\{duplicate, neutral\}$. For duplicate question detection, the ultimate goal is to optimize the standard binary cross-entropy loss function.

## 4 Experiment Setting

### 4.1 Datasets

We introduce two datasets covering fifteen different domains to evaluate our method. The first dataset is CQADupStack, and the second dataset comes from the Stack Overflow data dump. Table 1 shows the statistics of two datasets.

Table 1: The statics of datasets

| | CQADupStack | SODup |
|---|---|---|
| # of question pairs | 48090 | 100000 |
| # of intents | 259588 | 467816 |
| # of non-intents | 404373 | 595488 |
| intent rate | 39.1% | 44.0% |
| average intents per question | 3.12 | 3.56 |

**CQADupStack dataset** was released by Ahasanuzzaman et al. (2016) and has become a popular benchmark for CQA research (Nakov et al.,

2017). The dataset includes data from 12 CQA sites in Stack Exchange, such as Android, English, Game, Mathematics, Physics and Unix. We use duplicate questions and neutral questions in the dataset and randomly split them into training, validation, and test sets with a ratio of 8:1:1. However, the dataset does not include the largest forum Stack Overflow (SO) as the authors stated that SO data was too large to process at that time. To overcome the limitation, we have constructed an additional dataset, the Stack Overflow duplicate questions dataset (SODup).

**SODup dataset**. Stack Overflow is a widely used CQA forum that focuses on code-related questions. All questions and question links are available at the official data dump.[1] We downloaded the data dump and then filtered questions according to the three most popular programming languages: JavaScript, Python and Java. We extract the duplicate/neutral question relations based on LinkTypeId field of PostLinks table: (1) Duplicate relation: If a question pair has LinkTypeId=3, it means that the pair is marked as "Duplicate" by forum users. Therefore, we recognize the question pairs having LinkTypeId=3 as duplicate relations. (2) Neutral relation: Instead of just using random question pairs which are too simple for classification, we also gather indirectly referenced questions as neutral relations which serve as hard neutrals. Specifically, a question pair with LinkTypeId=1 means that one question references the other. However, some of the question pairs with a LinkTypeId=1 in the data dump are indeed duplicates in reality due to the delay of user annotation. To increase the probability of being true neutrals, we use questions with indirect references as neutral relations. The ratio of random and indirectly referenced questions is 1:1 ,and the ratio of duplicate and neutral relations is also 1:1. We adopt the same setting 8:1:1 for training, validation and test split in the SODup dataset.

### 4.2 Training Details

In order to avoid label leakages, there is **no overlap on question_ids** between the test dataset for duplication question detection and the dataset for intent recognition and matching. We train intent recognition and duplicate question detection with a batch size of 32. The batch size for contrastive learning in intent matching is 16. We set a maximum

---

[1] https://archive.org/details/stackexchange

Table 2: Overall performance of duplicate question detection.

| | SODup | | | CQADupStack | | | | | | | | | | | |
|---|---|---|---|---|---|---|---|---|---|---|---|---|---|---|---|
| | Js | Py | Java | And. | Eng. | Game | Gis | Math | Phys. | Prog. | Stats | Tex | Unix | Web. | Word. |
| BiMPM | 0.847 | 0.870 | 0.878 | 0.892 | 0.882 | 0.908 | 0.766 | 0.820 | 0.913 | 0.850 | 0.819 | 0.886 | 0.860 | 0.921 | 0.818 |
| MFAE | 0.853 | 0.869 | 0.890 | 0.901 | 0.870 | 0.857 | 0.758 | 0.832 | 0.899 | 0.876 | 0.818 | 0.897 | 0.894 | 0.917 | 0.784 |
| SBERT | 0.868 | 0.873 | 0.887 | 0.901 | 0.911 | 0.888 | 0.797 | 0.826 | 0.924 | 0.890 | 0.802 | 0.894 | 0.892 | 0.932 | 0.804 |
| SRoBERTa | 0.896 | 0.891 | 0.896 | 0.933 | 0.893 | 0.923 | 0.820 | 0.803 | 0.929 | 0.902 | 0.852 | 0.901 | 0.904 | 0.942 | 0.838 |
| Intent-DQD - Bert | 0.887 | 0.882 | 0.893 | 0.936 | 0.925 | 0.910 | 0.837 | **0.880** | 0.944 | 0.913 | **0.885** | 0.898 | 0.906 | 0.946 | 0.838 |
| Intent-DQD - Roberta | **0.908** | **0.902** | **0.914** | **0.956** | **0.925** | **0.956** | **0.838** | 0.874 | **0.946** | **0.931** | 0.852 | **0.921** | **0.921** | **0.953** | **0.878** |

sequence length of 512 for a single question and 128 for a single intent. We train 20 epochs for contrastive learning and 10 epochs for classification tasks. We set the learning rate to 5e-5 and used a training scheduler with a weight decay rate of 0.01 and one epoch warm-up. In intent recognition, we use Spacy package to extract POS information. For duplicate question detection, the top-3 performing checkpoints on the validation set will be evaluated on the test set to report average results.

# 5 Result and Analysis

## 5.1 Duplicate Question Detection Result

We compare Intent-DQD with a few baselines, including BiMPM(Wang et al., 2017), and MFAE(Zhang et al., 2020) that are representative duplicate question detection methods, as well as general-purposed SBERT and SRoBERTa models with fine-tuning. BiMPM encodes each question individually and then employs a multi-perspective matching mechanism for question pairs. MFAE utilizes pre-trained language models to encode questions and enhance keywords in question matching. Additionally, to compare with SBERT and SRoBERTa, we implement a BERT-based and a RoBERTa-based Intent-DQD, respectively.

Table 2 demonstrates the F1-score of classification results on fifteen domains. From the results, our Intent-DQD outperforms the baselines in all fifteen domains. The largest improvement is on the Math domain at 4.8%, and the least increase is 1.1% on the Python domain. We observe that even incorporating keyword emphasis on top of BERT, MFAE is not able to surpass general-purposed SBERT in most domains. One possible reason is that MFAE

does not differentiate intent, and all emphasized keywords can also appear in non-intent sentences, which may introduce extra noise when detecting duplicates.

Table 3: The result of ablation study. It shows the overall duplicate question detection performance on the two datasets. A, B and C donate three combinations of intent recognition (**ir**), intent matching(**im**) and relation aggregation(**ra**) illustrated in Formula 1.

| | ir | im | ra | SODup | CQADup. |
|---|---|---|---|---|---|
| A | ✗ | ✗ | ✗ | 0.894 | 0.896 |
| B | ✓ | ✗ | ✗ | 0.903 | 0.912 |
| C | ✓ | ✓ | ✗ | 0.900 | 0.914 |
| Intent-DQD | ✓ | ✓ | ✓ | 0.908 | 0.921 |

## 5.2 Ablation Study

Since Intent-DQD has different components, it is essential to dissect the impact of each component contributing to performance improvement. Considering that Roberta-based Intent-DQD achieves the highest performance in most forums, we choose it for the following ablation experiments.

As shown in Table 3, the combination A is a SRoBERTa-only model which performs the worst, and the complete Intent-DQD achieves the highest performance on both datasets. With only intent recognition, combination B can only feed the recognized intents into Bi-Encoder and combine encoded intents with encoded questions. Even so, we see an increase in performance, which suggests the significance of utilizing intents for detecting duplicate questions. However, in combination C, only using intent matching can slightly increase performance on CQADupStack dataset and even decrease

| | ir | im + ra | Intent-DQD | SRoBERTa |
|---|---|---|---|---|
| Case **1**: Duplicate | QA: Difference between "I have got" and "I have gotten"? I see these two expressions are used almost identically in different contexts. Is there a difference between I have got and I have gotten?
QB: I notice that Americans use the word 'gotten' when we in Britain just use 'got', is 'gotten' accepted American English, that is, used and accepted in English examination papers, or is it a type of slang, used in speech, but not written? The previous answer you directed me to doesn't quite answer my question, although it was interesting. I can deduce from it. However, that 'gotten', if used correctly, is indeed acceptable and an actual word, is that right? | $\begin{array}{c c c} & I_1^B & I_2^B \\ I_1^A & .72 & .43 \\ I_2^A & .77 & .41 \end{array}$ $\begin{array}{c c c} & I_1^A & I_2^A \\ I_1^A & 1 & .92 \\ I_2^A & .92 & 1 \end{array}$
$\begin{array}{c c c} & I_1^B & I_2^B \\ I_1^B & 1 & .56 \\ I_2^B & .56 & 1 \end{array}$ ⇨ $\begin{array}{c c c} & I_1^B & I_2^B \\ I_1^A & .58 & .22 \\ I_2^A & .62 & .21 \end{array}$ | Duplicate | Neutral |
| Case **2**: Neutral | **QA**: Python question How can I loop back my code? Currently I have my code exiting when an error is detected. It will exit if your income is inputted as a negative and will exit if your postcode is entered as a string. But I need to be able to create a loop so it will give you another chance to re-enter your values without error exiting. New to python and haven't figured it out. Any help would be very appreciated :)
**QB**: How to loop back to specific point in code. So I'm programming a litlle game and I'm trying to do something which I don't understand how to do. I've defined a function, and when none of the conditions with which the code works, I want it to go back to an other line of code. Please be as clear as possible when answering because I've just started to learn programming and I know almost nothing about it. | $\begin{array}{c c c} & I_1^B & I_2^B \\ I_1^A & .62 & .45 \\ I_2^A & .29 & .13 \end{array}$ $\begin{array}{c c c} & I_1^A & I_2^A \\ I_1^A & 1 & .54 \\ I_2^A & .54 & 1 \end{array}$
$\begin{array}{c c c} & I_1^B & I_2^B \\ I_1^B & 1 & .57 \\ I_2^B & .57 & 1 \end{array}$ ⇨ $\begin{array}{c c c} & I_1^B & I_2^B \\ I_1^A & .49 & .45 \\ I_2^A & .39 & .35 \end{array}$ | Neutral | Duplicate |

Figure 4: Two cases that Intent-DQD correctly makes predictions while SRoBERTa makes wrong predictions

the performance on SODup dataset. This is likely due to the fact that questions in forums tend to have multiple intents, and without effectively using intent-level relations, it is not comprehensive enough to determine question-level relations. As shown in Table 1, SODup has higher average intent per question than CQADupStack, thus more challenging without proper question-level aggregation. Lastly, we observe that aggregating intent-level relation can significantly improve classification performance, which further verifies that the relation aggregation module is crucial and it can only achieve its effectiveness when paired with the intent matching module.

### 5.3 Intent Recognition Verification

Since the intent recognition is the basis of our method, it is essential to verify the performance of our intent recognizer. We randomly sample 5,000 sentences from QA forums and manually annotate whether a sentence is intent or not. After reaching an agreement ($kappa > 0.9$), authors annotated a test dataset including 1,916 sentences labeled as $intent$ and 3,084 labeled as $non\text{-}intent$. We then verify our intent recognition module on the test dataset. Table 4 shows that either regex or title strategy is valuable for recognizing intents. The extra improvement made when combining regex and title suggests that they are complementary to each other. Lastly, the F1-score reaching 0.847 with structure features verifies the effectiveness of all sub-modules within our intent recognizer.

Table 4: As mentioned in Sec 3.1, **R** uses Regex patterns to find intents for train data. **T** treats Titles as intents for train data. **S** integrates sentence Structure features(POS, Sentence Position, and Keyword) in training. Our result in table 4 shows that the combination of **R**, **T** and **S** can produce an effective intent recognizer without human annotation.

| | Precision | Recall | F1 |
|---|---|---|---|
| R | 0.816 | 0.621 | 0.706 |
| T | 0.914 | 0.633 | 0.748 |
| R+T | 0.862 | 0.809 | 0.834 |
| R+T+S | 0.864 | 0.831 | 0.847 |

### 5.4 Case Study

To better understand how Intent-DQD works, we have studied two test cases from the CQADupStack and SODup datasets. To analyze the contributions of each component, we further provide intermediate outputs to illustrate how intent recognition, intent matching, and duplicate question detection contribute to duplicate detection in the two cases.

The first case consists of two questions from the English domain. The label of the question pair is "Duplicate" as both questions are asking for the difference between "got" and "gotten". The prediction of intent-DQD is "Duplicate" while SRoBERTa predicts "Neutral". There are two main reasons for SRoBERTa's mistake: (1) The non-intent content has a semantic discrepancy. (2) The intent content between the two questions has a large lexical gap. With the help of contrastive learning, our intent

matching component in Intent-DQD is able to output a high matching score of the intent pairs, thus closing the lexical gap. As seen in Figure 4, QA has quite consistent intents, and the main intent of QB matches the intent of QA.

The second case consists of two questions from the python domain. The label of the question pair is "Neutral". Although both questions are about looping in python, the nuances of the intents make a difference i.e. the second intent of $Q_A$ is to catch errors and loop, while the second intent of $Q_B$ is to loop after condition control such as if-else. Additionally, the non-intents of the two questions are similar, and they both indicate that the question authors have little experience in the python domain. Therefore, it is not surprising that SRoBERTa predicts it as "Duplicate". As for Intent-DQD, the intent matching component can discover the difference between $I_2^A$ and $I_2^B$ because of training from hard negative samples. In the end, the relation aggregation component can comprehensively consider intent-level relations and produce relatively low question-level match scores for the duplication classification.

### 5.5 Scalability in Real-world

In our method, given a new question, the intermediate results of intent recognition and intent embeddings can be stored and reused. Additionally, a scalable paradigm for detecting duplicates typically involves two stages: (1) a coarse-grained stage that retrieves duplication candidate question pairs and (2) a fine-grained stage that classifies candidates to detect duplication. Intent-DQD contributes directly to the fine-grained stage and thus our evaluation focuses on the classification results. As for the coarse-grained stage, our intent embeddings can be utilized to construct question-level embeddings and extend to duplication candidate retrieval.

## 6 Related Work

Instruction tuning is commonly regarded as the first step in the RLHF training process for training human-aligned LLMs including GPT-3.5, GPT-4, Claude (Ouyang et al., 2022; Bai et al., 2022). The goal of instruction tuning is to train LLM in following human intents expressed in instructions (OpenAI, 2023; Taori et al., 2023; Chiang et al., 2023). For instance, when a human asks a question to a LLM, their intent is to receive a proper answer. Before instruction tuning, a pre-trained LLM may

continue the question or generate similar questions, whereas an aligned LLM trained through instruction tuning is more likely to generate an answer based on the question.

Collecting high-quality training data is fundamental for effective instruction tuning. CQA forums have millions of well-aligned Q&A data, making them ideal sources for instruction tuning (Zhou et al., 2023). However, the issue of duplicate questions arises as the volume of content within CQA continues to grow, posing a threat to content quality. Recent research highlights the importance of detecting and removing duplication to optimize for LLM instruction tuning. Hernandez et al. (2022) find deduplication increases generalization ability and learning stability. It can also save computation and decrease train-test set overlap Lee et al. (2022). Deduplication also reduces privacy risks because LLM is vulnerable to privacy attacks when sensitive information is duplicated in the instruct-tuning dataset (Kandpal et al., 2022).

Existing duplication detection studies mainly build text-pair classifiers based on bi-encoders or cross-encoders. Bi-encoders (Mueller and Thyagarajan, 2016; Conneau et al., 2017; Reimers and Gurevych, 2019; Zhang et al., 2020), encode each question individually while cross-encoders (Devlin et al., 2019; Liang et al., 2019; Liu et al., 2019; He et al., 2020), use cross-attention between questions or self-attention by concatenating multiple questions. Despite the popularity of cross-encoders, they typically have high computational cost (Reimers and Gurevych, 2019) and inconsistent performance (Chen et al., 2020). In this work, we focus on bi-encoders, which are more computationally efficient and have shown consistent performance in duplicate question detection. However, the above-mentioned methods are generally generic text-pair matching and do not consider intents in question, which can lead to sub-optimal results in deduplicating datasets for instruction tuning.

Therefore, we incorporate intent features in Intent-DQD to detect duplicate questions. Unlike existing research on intents (Kim et al., 2016; Coucke et al., 2018; Weld et al., 2021, 2022) which typically rely on human annotation, our proposed method is annotation-free, which leverages characteristics of CQA forums and automatically mine labels for intents.

## 7   Conclusion

In this work, we propose an intent-based duplicate question detection method Intent-DQD. We design intent recognition to identify question intent without human annotation. Intent matching employs contrastive learning to enhance the embeddings. Duplicate question detection effectively aggregates intent-level relations and obtains question-level relations. Evaluation on fifteen domains from two real-world datasets shows that our method outperforms existing state-of-the-art methods.

## 8   Limitations

The main assumption of this work is that the user-generated content in CQA forums contains explicit intent. The effectiveness of our carefully designed intent recognition, intent matching and duplicate detection modules may have limited success if the questions do not have clear intent. This can be true in forums that focus on open discussions, where the objective of the discussion is unclear.

## 9   Ethics Statement

We acknowledge and conform ACL and ACM Code of Ethics. The datasets we used are from the official public data source, and we use the data under the license. In data processing, we do not use user private information. Our method does not create toxic content or bias in different groups.

## 10   Acknowledgments

We would like to thank the anonymous reviewers and appreciate their insightful comments that helped improve the quality of the paper. This work was supported by the National Natural Science Foundation of China (NSFC) under Grants 61932007 and 62172106. Peng Zhang is a faculty of School of Computer Science, Fudan University. Tun Lu is a faculty of School of Computer Science and Shanghai Key Laboratory of Data Science, Fudan University and Shanghai Institute of Intelligent Electronics & Systems. Peng Zhang and Tun Lu are the corresponding authors.

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

# A  Appendix

## A.1  Regex Expression Examples

- Regex expressions for mining intents:

```
does( [^\s]+){0,3} work
can (I|we) (use|apply|detect)
(tell|explain)( [^\s]+){0,3} why
(does|know|do)( [^\s]+){0,5} (exist)
(is)( [^\s]+){0,5} (a|any|some)( [^\s]+){0,3} (way)
how (can|could|should|do|does) (I|you)( [^\s]+){0,10}\?
((re)?solve|fix)( [^\s]+){0,5} (error|(\w{0,35})exception)
(errors?|exceptions?)( [^\s]+){0,5} (logcat|stacktrace|log|message)
(log\s?cat|stack\s?trace|log|gradle)( [^\s]+){0,5} (error|exception)
```

- Patterns for removing noise:

```
how( [^\s]+){0,5} solve (it|this|that)+
understand( [^\s]+){1,5} how ( [^\s]+){0,5} (it|this|that)+
(try|tried|trying)( [^\s]+){1,5} how ( [^\s]+){0,5} (it|this|that)+
```

## A.2 Detail Statics of Datasets

Table 5 shows the statics of each domain in SODup dataset, and Table 6 shows the CQADupStack dataset.

Table 5: The statics of SODup dataset.

| Domain | # of Question Pairs | # of Intents |
|---|---|---|
| Javascript | 33334 | 171061 |
| Python | 33334 | 153704 |
| Java | 33334 | 143051 |

Table 6: The statics of CQADupStack dataset.

| Domain | # of Question Pairs | # of Intents |
|---|---|---|
| Android | 3426 | 18529 |
| English | 7782 | 32412 |
| Game | 4558 | 23813 |
| Gis | 2232 | 14270 |
| mathematics | 2744 | 15793 |
| Physics | 3936 | 22075 |
| Programmers | 3476 | 22315 |
| Stats | 1832 | 11714 |
| Tex | 10390 | 54887 |
| Unix | 3428 | 20164 |
| Webmasters | 2790 | 14474 |
| Wordpress | 1496 | 9142 |

## A.3 The Effect of Contrastive Learning

We randomly sample a batch of matched intents and mismatched intents. We visualize the similarity between intents in two situations: (1) without contrastive learning(CL), we encode intents through pre-trained BERT (2) with contrastive learning, we further train the BERT through the contrastive loss function. As shown in Figure 5, after contrastive learning, matched intents (diagonal elements) have a more prominent distinction with mismatched intents (off-diagonal elements).

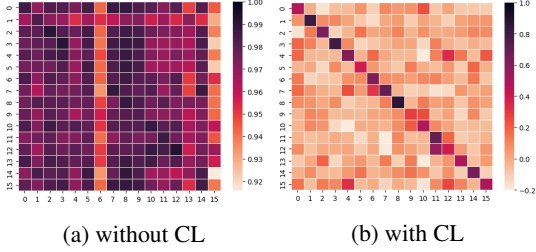

(a) without CL          (b) with CL

Figure 5: Comparison of intent cosine similarity before or after contrastive learning. The diagonal elements are matched intent pairs, and the non-diagonal elements are mismatched intent pairs. The color of the cell indicates the similarity score.

## A.4 ChatGPT for Duplicate Question Detection

We have randomly selected 100 question pairs from SODup test data and 100 pairs from CQADupStack test data. Then we ask ChatGPT (GPT-3.5-turbo) to classify these pairs, and the final F1-scores are 0.855 and 0.861 for SODup and CQADupStack, whereas the F1-scores for Intent-DQD are 0.9 and 0.92 respectively. The results imply that duplicate question detection can be more complex than initially anticipated. The complexity arises from the inherent difficulty in precisely defining what constitutes duplication. In CQA forums, duplicate questions are not strictly limited to having exact lexical or semantic equality. Instead, duplication can encompass scenarios where one question can be inferred from or infer another question to some degree. However, it is difficult to have an explicit definition, making it equally challenging for Chat-GPT to accurately detect duplication.

## A.5 Relationship between the intent recognition and DQD

Table 7 presents an extended experiment examining the relationship between the accuracy of intent recognition and the performance of duplicate question Detection(DQD). The table indicates that a higher F1 score in intent recognition correlates with a higher F1 score in DQD.

Table 7: **R** employs Regex patterns. **T** interprets titles as intents. **S** incorporates sentence structure features, including Part-of-Speech (POS), sentence position, and keyword identification. The first two columns display the performance of detecting duplication in SODup and CQADup, respectively, while the final column showcases the F1-score for intent recognition.

| | SODup F1 | CQADup F1 | Intent F1 |
|---|---|---|---|
| R | 0.861 | 0.879 | 0.706 |
| T | 0.875 | 0.892 | 0.748 |
| R+T | 0.904 | 0.916 | 0.834 |
| R+T+S | 0.908 | 0.921 | 0.847 |

The rank of DQD performance across various strategies is consistent with the rank of the F1 score for intent recognition. This correlation can be attributed to the phenomenon of error propagation: inaccuracies in intent recognition can lead to subsequent mistakes in intent matching, which in turn influence the final DQD results.