# OpenReview forum: "An Intent-based and Annotation-free Method for Duplicate Question Detection in CQA Forums"
_EMNLP/2023/Conference — EMNLP 2023 Findings_

### Official Review · Reviewer_xaBD · 2023-08-05

**Soundness:** 3

**Excitement:**

3: Ambivalent: It has merits (e.g., it reports state-of-the-art results, the idea is nice), but there are key weaknesses (e.g., it describes incremental work), and it can significantly benefit from another round of revision. However, I won't object to accepting it if my co-reviewers champion it.

**Missing References:**

Paper doesn’t discuss the following significant works in the same direction:

1. Divide and Conquer: Text Semantic Matching with Disentangled Keywords and Intents.

2. Match2: Matching over Matching Model for Similar Question Identification

**Paper Topic And Main Contributions:**

The paper focuses on leveraging intent information to detect duplicate question on Community Question Answering (CQA) forums. These CQA forums can serve as a high-quality knowledge bases for instruction tuning for LLMs, which in turn, can be employed in designing systems like ChatGPT etc.

The proposed model, Intent-DQD, extracts abundant training labels to recognize and match intents without human annotations. The authors then utilize supervised contrastive learning to match semantically similar intent pairs and separate the non-identical ones. They further propose a method to aggregate intent-level relations and obtain question-level relations to detect duplicate questions with multiple intents. They experimented on two datasets covering 15 domains to show the effectiveness of the proposed method.

**Questions For The Authors:**

What’s the ration of positive and negative samples in intent matching module? And does it affect the training of the module?

**Reasons To Accept:**

The paper contributes towards one of the critical challenges in the field of NLP i.e. detecting and eliminating duplicate content from the CQA forums to improve the content quality.

Tackles the challenge of missing labels by utilizing regular expression patterns. The proposed method is annotation-free which helps in saving the cost of labeling.

Handles multi-intent questions by exploiting intent-level relations.

The proposed model is scalable in real-world scenarios.

**Reasons To Reject:**

Requires intent to be explicitly mentioned in the title/body. Doesn’t work well where intents are implicit.

The paper refers to the arxiv references when the conference references exist such as line 663, 639, etc.

The paper has typo and grammar mistakes which hinders the readability of the paper such as missing commas (line 185), misformed sentence (line 166), extra hyphens (line 158), combining (line 440), and missing equation number (line 290, 293) etc.

**Reproducibility:**

4: Could mostly reproduce the results, but there may be some variation because of sample variance or minor variations in their interpretation of the protocol or method.

**Reviewer Confidence:**

5: Positive that my evaluation is correct. I read the paper very carefully and I am very familiar with related work.

---

> ### Author Rebuttal · Authors · 2023-08-29
>
> ### Response 1:
> > Requires intent to be explicitly mentioned in the title/body. Doesn't work well where intents are implicit.
>
> Our method handles implicit intents. Compared with an explicit intent in the title or body, such as "How to solve GPU OOM error in Pytorch?", an implicit one can be "I have GPU OOM error in Pytorch", which implicitly indicates that the user needs a solution to solve the OOM error. As mentioned in section 5.3, we have conducted verification of intent recognition through manually annotated data. Among the 1,916 labeled intents, we further identify 189 intents belonging to implicit intents similar to the above example. We then evaluate the intent recognizer, and the F1 is 0.833. It is comparable to the F1 of 0.847 for all the explicit and implicit intents, showing the intent recognition module is able to detect implicit intent.
>
> &emsp; However, our method does not work well on open conversations with undefined intents. Examples such as "I love Python and it is an amazing language" can confuse answerers on what the question is asking for and are commonly considered low-quality questions, which are marked with negative scores and not considered useful in CQA forums. We will revise the limitations section to reflect the above data points and clarifications.
>
> &nbsp;
>
> ### Response 2:
> > The paper refers to the arxiv references when the conference references exist such as line 663, 639, etc.
> Thank you for pointing this out and we will update the references.
>
> &nbsp;
>
> ### Response 3:
> > The paper has typo and grammar mistakes which hinders the readability of the paper such as missing commas (line 185), misformed sentence (line 166), extra hyphens (line 158), combining (line 440), and missing equation number (line 290, 293) etc.
> Thank you and we will fix the grammar mistakes and improve readability.
>
> &nbsp;
>
> ### Response 4:
> > What's the ratio of positive and negative samples in intent matching module? And does it affect the training of the module?
>
> The ratio of positive and negative samples is 1:1. The reason behind this is that there are two equally important goals as follows:
>  * Goal_A: decreasing the distance among matched intents
>  * Goal_B:  increasing the distance among mismatched intents.
>
> &emsp; The positive/negative ratio indeed affects the training of the intent matching module. We conducted training on the ratio of 2:1 and 1:2. We observed that a lower proportion of positive samples or negative samples can cause sub-optimal performance on either Goal_A or Goal_B. Therefore, we set the ratio of matched intents and mismatched intents to 1:1 during contrastive learning. We will incorporate this into the paper's final version to explain the design choices.

---

### Official Review · Reviewer_118e · 2023-08-05

**Typos Grammar Style And Presentation Improvements:** Figure 1
**Soundness:** 3

**Excitement:**

3: Ambivalent: It has merits (e.g., it reports state-of-the-art results, the idea is nice), but there are key weaknesses (e.g., it describes incremental work), and it can significantly benefit from another round of revision. However, I won't object to accepting it if my co-reviewers champion it.

**Paper Topic And Main Contributions:**

The paper targets the duplicate question problem in CQA forums.  It particularly focuses on leveraging question intents to check the duplication.  The work aims to have more accurate duplicate detection.

**Questions For The Authors:**

Pls see "Reasons To Reject"

**Reasons To Accept:**

1. It targets an interesting and useful topic as the CQA forum is not just for human users to find solutions, but also is leveraged to pret-train language models.
2. The solution was tested effective and outperforms many pre-train models

**Reasons To Reject:**

1. It is unclear how the intents are extracted from section 3.1.
2.Figure 3 has issues: the out put of the intent recognizer is the binary label (b), however, in (a), it should be the intents.  It is not clear how the two parts are connected.
3. The intent recognizer has a transformer-based encoder, which will dominate the features-- i.e. making the three features (tag, position, POS) show less importance. Also, experiments on the effectiveness of the three features are needed.
4. The writing also has issues. For example, before line 180, the method does not mention contrastive learning.  So it is better to first have a overall method framework introduced, then talk about the details.
 5. Inter-link depends on the existing duplicate links provided, so it is not a generic method.

**Reproducibility:**

5: Could easily reproduce the results.

**Reviewer Confidence:**

5: Positive that my evaluation is correct. I read the paper very carefully and I am very familiar with related work.

---

> ### Author Rebuttal · Authors · 2023-08-29
>
> ###  Response 1:
> > It is unclear how the intents are extracted from section 3.1. 2.Figure 3 has issues: the output of the intent recognizer is the binary label (b), however, in (a), it should be the intents. It is not clear how the two parts are connected.
>
> Sorry for the confusion in Figure 3. The process of how we extract intents and how part (a) and part (b) are connected are as follows :
>  * We first use an intent recognizer to predict the binary label (Intent or Not-intent) for each sentence.
>  * After collecting all predictions, we filter sentences with Intent labels as intents.
>  * These intents will passed to the encoder of the intent matcher. Here, we encode each intent into a real number vector.
>  * The intent matcher calculates the similarity of vectors. A higher similarity between intents indicates a higher possibility of being matched.
>
> &emsp; Thanks for your advice on Figure 3. We will add the "filtering by Intent prediction" for clarity.
>
> &nbsp;
>
> ###  Response 2:
> > The intent recognizer has a transformer-based encoder, which will dominate the features-- i.e. making the three features (tag, position, POS) show less importance. Also, experiments on the effectiveness of the three features are needed.
>
> Adding extra features to a transformer-based encoder can refer to existing research: (1) The work [1]  proves that the feature indicating whether a token belongs to a code or an entity can improve the performance of the NER task in code-mixed text. (2) The embedding of sentence position can be considered as the extension of segment embedding in transformer-based models like BERT. The segment embedding is effective for segment-sensitive tasks such as next-sentence prediction and sentence pair classification. In our study, since we observed that some users prefer to put intents at the start or the end of a post and other usage of tags and POS features, we try to incorporate the extra tag, position, and POS features for best prediction results.
>
> &emsp; **Experiments of the three features.** The individual POS, Keyword and SentencePOS boost F1 from 0.834 (no structure features) to 0.844, 0.843 and 0.839, respectively. Table 4 demonstrates that we increase the intent recognition F1 from 0.834 to 0.847 with all structure features. The overall duplicate question detection F1 increases from 0.916 to 0.921 on CQADup with the three features.
>
> *[1] Tabassum J, Maddela M, Xu W, et al. Code and Named Entity Recognition in StackOverflow[C]//Proceedings of the 58th Annual Meeting of the Association for Computational Linguistics. 2020: 4913-4926.*
>
> &nbsp;
>
> ###  Response 3:
> > The writing also has issues. For example, before line 180, the method does not mention contrastive learning. So it is better to first have an overall method framework introduced, then talk about the details.
>
> Thanks for your advice, and we will improve the method section. We will add a concise description of the technique of each submodule, especially describing the role of contrastive learning in intent matching.
>
> &nbsp;
>
> ###  Response 4:
> > Inter-link depends on the existing duplicate links provided, so it is not a generic method.
>
> We would like to clarify that inter-links (existing duplicate links) are used in training only. They are not used in inference. During the training process, existing duplicate links are used as both the labels for the final duplicate question detection and also part of the contrastive loss in the intent matching module. Intent-DQD, as well as all the baseline methods for CQA duplicate detection we compared in the paper, relies on existing duplicate links as training labels.
>
> &emsp; There are sufficient existing duplicate links in CQA forums for training. Currently, multiple CQA forums provide human-voted duplicate links. For instance, the number of duplicate links in the Stack Overflow forum has reached 1,453,083. Votes are of high quality because only experienced users with a reputation >= 3000 can vote for duplicate links, and at least three votes to duplicate can finally create a duplicate link.
>
> &emsp; To further test the situation where a new forum has just formed and has no existing duplicate link for training, we conduct a cold-start experiment: training Intent-DQD on SODup data and testing it on CQADup. The overall F1 score of intent-DQD reaches 0.806, compared to SRoBERTa in the same cold-start situation with an F1 of 0.757. This indicates the generalization of intent-DQD in the situation where it cannot use any existing duplicate link for training in a new forum.
>
> &nbsp;
>
> ###  Response 5:
> > Figure 1: Q1 and Q2 should have matched intent, not mismatch
>
> The involving parts of our paper are:
>  * In Figure 1, the highlight means the intent and Q1 and Q2 indeed have matched the intent.
>  * In the caption of Figure 1, our description is "Q1 and Q2 have matched intent to solve a particular java error while having mismatched backgrounds."
>
> &emsp; Therefore, we indeed express a consistent meaning that Q1 and Q2 have matched the intent but mismatched background information. We will make the description more clear in the final version.

---

### Official Review · Reviewer_Q6eK · 2023-08-05

**Soundness:** 4

**Excitement:**

4: Strong: This paper deepens the understanding of some phenomenon or lowers the barriers to an existing research direction.

**Paper Topic And Main Contributions:**

The authors propose an intent-based and annotation-free method (Intent-DQD) for duplicate question detection in CQA forums. The Intent-DQD method works in three steps: intent recognition, intent matching, and duplication detection. They leverage the characteristics of CQA forums to automatically extract training data for recognizing and matching intents. Experiments on datasets from two CQA forums demonstrate the effectiveness of the proposed method.

**Questions For The Authors:**


1. As intent recognization plays a key role in the whole process, how the different strategies (R, T, R+T) for extracting training data to learn an intent recognizer affect the overall performance of DQD?

2. How can we apply the Intent-DQD method in a new CQA forum?

**Reasons To Accept:**


1. An intent-based and annotation-free method for duplicate question detection in CQA forums is proposed. It detects duplicate questions based on intents they have, and does not require human annotation for preparing training data.

2. Extensive experiments on 15 domains from two real-world datasets demonstrate the effectiveness of Intent-DQD.

3. The paper is clearly written.

**Reasons To Reject:**


1. It's not clear whether the proposed method can be easily applied in a new CQA forum, as regex patterns may not be always valid across different forums.

**Reproducibility:**

4: Could mostly reproduce the results, but there may be some variation because of sample variance or minor variations in their interpretation of the protocol or method.

**Reviewer Confidence:**

4: Quite sure. I tried to check the important points carefully. It's unlikely, though conceivable, that I missed something that should affect my ratings.

**Typos Grammar Style And Presentation Improvements:**


Typos:

1. lines 072 and 073, missing parentheses around the citations
2. in the caption of figure 5, "before of after"

---

> ### Author Rebuttal · Authors · 2023-08-29
>
> ### Response 1:
> > As intent recognization plays a key role in the whole process, how the different strategies (R, T, R+T) for extracting training data to learn an intent recognizer affect the overall performance of DQD?
>
> We have observed that a higher F1 of intent recognition results in a higher F1 of duplicate question detection (DQD) as shown in the table below:
>
> |      |SODup F1|CQADup F1| Intent F1|
> | ---- | ---- | ---- | -------- |
> |   R     |0.861 |0.879 | 0.706    |
> |   T     |0.875 |0.892 | 0.748    |
> |   R+T   |0.904 |0.916 | 0.834    |
> |   R+T+S |0.908 |0.921 | 0.847    |
>
> &emsp; The rank of the overall performance of DQD with different strategies follows the rank of F1 for intent recognition. One possible reason is the error transition. Errors in intent recognition will cause subsequent errors in intent matching, and then they can be reflected in the final DQD.
>
> &nbsp;
>
> ### Response 2:
> > How can we apply the Intent-DQD method in a new CQA forum?
>
> When applying the Intent-DQD method in a new CQA forum using the training data in the new forum, the main challenge is how to adopt the domain features of the new forum. Although the regex pattern has difficulties in handling unknown/new domain features (including specific text style and specific terms), the following key points in our method can help:
>
> * The question title is a part of the forum content. In intent-DQD, treating titles as intents can address partial domain features. (line 140)
> * The tag of questions usually collects specific items in a forum. To handle specific terms, we build a keyword library based on the tag, and mark each token whether it belongs to a keyword during intent recognition. (line 192)
> * We use POS to emphasize the structure feature. (line 204)
>
> &emsp; For a cold start situation where a new forum is just formed and has almost no data available for training, we train an intent recognizer on data from SODup and test it on data from CQADupStack. The intent recognition F1 is 0.814, which is comparable to the overall 0.847 F1. The overall deuplicate detection F1 is 0.806, compared to SRoBERTa in the same cold-start situation with an F1 of 0.757. It suggests that Intent-DQD is able to generalize to new CQA forums in a cold start situation using training data from old forums. As we build more data and knowledge in the new forum, e.g., regular expression patterns and human-labeled duplication links, we will be able to retrain Intent-DQD using training data from the new forum and achieve better performance.

---

### Meta-Review · Area_Chair_GbAr · 2023-09-19

**Recommendation:** 3

**Metareview:**

All the reviewers agree on the fact that the paper provides a valuable contribution to the research on Community Question Answering (CQA) forums. In this paper, an intent-based and annotation-free method for duplicate question detection in CQA forums is proposed, that detects duplicate questions based on the intents they have, and does not require human annotation for preparing training data. Extensive experiments on several domains from two real-world datasets have been carried out to demonstrate the effectiveness of the proposed method. However, some concerns have been raised by the reviewers concerning the complexity of applying such method to a new CQA forum, and the effectiveness of the intent extraction method in particular for implicit and undefined intents. The writing of the paper should also be improved.

---

### Decision · Program_Chairs · 2023-10-07

**Decision:**

Accept-Findings

**Comment:**

All the reviewers agree on the fact that the paper provides a valuable contribution to the research on Community Question Answering (CQA) forums. In this paper, an intent-based and annotation-free method for duplicate question detection in CQA forums is proposed, that detects duplicate questions based on the intents they have, and does not require human annotation for preparing training data. Extensive experiments on several domains from two real-world datasets have been carried out to demonstrate the effectiveness of the proposed method. However, some concerns have been raised by the reviewers concerning the complexity of applying such method to a new CQA forum, and the effectiveness of the intent extraction method in particular for implicit and undefined intents. The writing of the paper should also be improved.